# 3D Printed Hollow Microneedles for Treating Skin Wrinkles Using Different Anti-Wrinkle Agents: A Possible Futuristic Approach

**Humayra Islam** [1,†]**, Taslima Sultana Poly** [1,†]**, Zarin Tasnim Tisha** [1,†]**, Samia Rahman** [1,†]**, Ahmed Issa Jahangir Naveed** [1,†]**, Alifa Ahmed** [1,†]**, Saraf Nawar Ahmed** [1,†]**, Jasmin Hassan** [1,†]**, Md. Jasim Uddin** [1,2,3,*,†] **and Diganta B. Das** [4,*,†]

1  Drug Delivery & Therapeutics Lab, Dhaka 1212, Bangladesh
2  Faculty of Engineering and Science, University of Greenwich, Chatham Maritime, Kent ME4 4TB, UK
3  School of Pharmacy, Brac University, 66 Mohakhali, Dhaka 1212, Bangladesh
4  Department of Chemical Engineering, Loughborough University, Epinal Way, Loughborough LE11 3TU, UK
*  Correspondence: jasim.uddin@bracu.ac.bd (M.J.U.); d.b.das@lboro.ac.uk (D.B.D.)
†  These authors contributed equally to this work.

**Abstract:** Skin wrinkles are an inevitable phenomenon that is brought about by aging due to the degradation of scleroprotein fibers and significant collagen reduction, which is the fundamental basis of anti-wrinkle technology in use today. Conventional treatments such as lasering and Botulinum toxin have some drawbacks including allergic skin reactions, cumbersome treatment procedures, and inefficient penetration of the anti-wrinkle products into the skin due to the high resistance of stratum corneum. Bearing this in mind, the cosmetic industry has exploited the patient-compliant technology of microneedles (MNs) to treat skin wrinkles, developing several products based on solid and dissolvable MNs incorporated with antiwrinkle formulations. However, drug administration via these MNs is limited by the high molecular weight of the drugs. Hollow MNs (HMNs) can deliver a wider array of active agents, but that is a relatively unexplored area in the context of antiwrinkle technology. To address this gap, we discuss the possibility of bioinspired 3D printed HMNs in treating skin wrinkles in this paper. We compare the previous and current anti-wrinkling treatment options, as well as the techniques and challenges involved with its manufacture and commercialization.

**Keywords:** antiwrinkle agents; hollow microneedles; skin wrinkles; 3D printing

## 1. Introduction

Skin wrinkles, a dominating phenotype, are natural and the most common phenomenon of our aging process [1]. It is an involuntary biological process that is brought about by a series of cutaneous associated changes such as skin elasticity loss, thinning of the epidermis, and loss of fat volumes under the skin [2].

There are several extrinsic and intrinsic factors that play a key role in the development of wrinkles [3]. These factors are capable of altering both epidermal and dermal structures, and function significantly [4]. Extrinsic factors such as radiation, smoking habits, and environmental pollution can contribute to the aging process by altering the oxidative stress of the skin which results in affecting the cellular process negatively. Moreover, factors such as (i) diabetes mellitus, (ii) pathophysiological changes during menopause, (iii) metabolic illness in the elderly, and (iv) use of certain drugs such as corticosteroids, promote wrinkle formation internally. For example, diabetes mellitus causes glycation which is an indirect way of causing skin wrinkles. During menopause, the level of estrogen decreases affecting the skin components, which is responsible for changes in skin texture, xerosis, and dermal atrophy. Our unhealthy lifestyles, such as a lack of sleep and food habits without sufficient antioxidants, also play a vital role in our skin health [3,5].

Photoaging, a form of extrinsic aging, involves premature changes to skin health due to continuous sun exposure. UV radiation acts as the central driver in this case [6,7]. Clinical features of both intrinsic and extrinsic skin aging mostly exist in a superimposed manner, however, they are completely distinct [8,9]. Usually, intrinsic aging is distinguished by increased fragility, xerosis, loss of underlying fat, fine wrinkles, and pruritus [4]. Contrastingly, extrinsic aging exhibits flimsy skin phenotypes including skin dullness, fine and coarse wrinkles, less atrophy, roughness, mottled pigmentation, and telangiectasias [10]. In addition, a couple of more characteristics of extrinsic aging are leathery skin, deeper rhytid, and dyschromia [11,12].

Wrinkles can be classified into four types depending on their anatomical characteristics, i.e., atrophic, elastotic, expressional, and gravitational wrinkles. Expressional wrinkles have well-defined patterns and lateral canthal rhytids popularly known as Crow's feet, which is an ideal example of skin wrinkles [13]. Variation in wrinkles also depends on heredity. MC1R gene has been found to have an influence on photoaging and spots, although its direct influence on skin wrinkles has not been found yet [1]. In addition, as mentioned earlier, our skin is continuously exposed to light, pollution, and radiation, causing gradual physiological alterations in skin layers, which makes the skin turnover and cell cycle slow and decreases the production of collagen VII as well. The loss of fibrillin-positive structures and reduction of collagen type VII mainly contribute to the formation of wrinkles as they weaken the bond between the dermis and epidermis of aged skin (see more in Figure 1) [14].

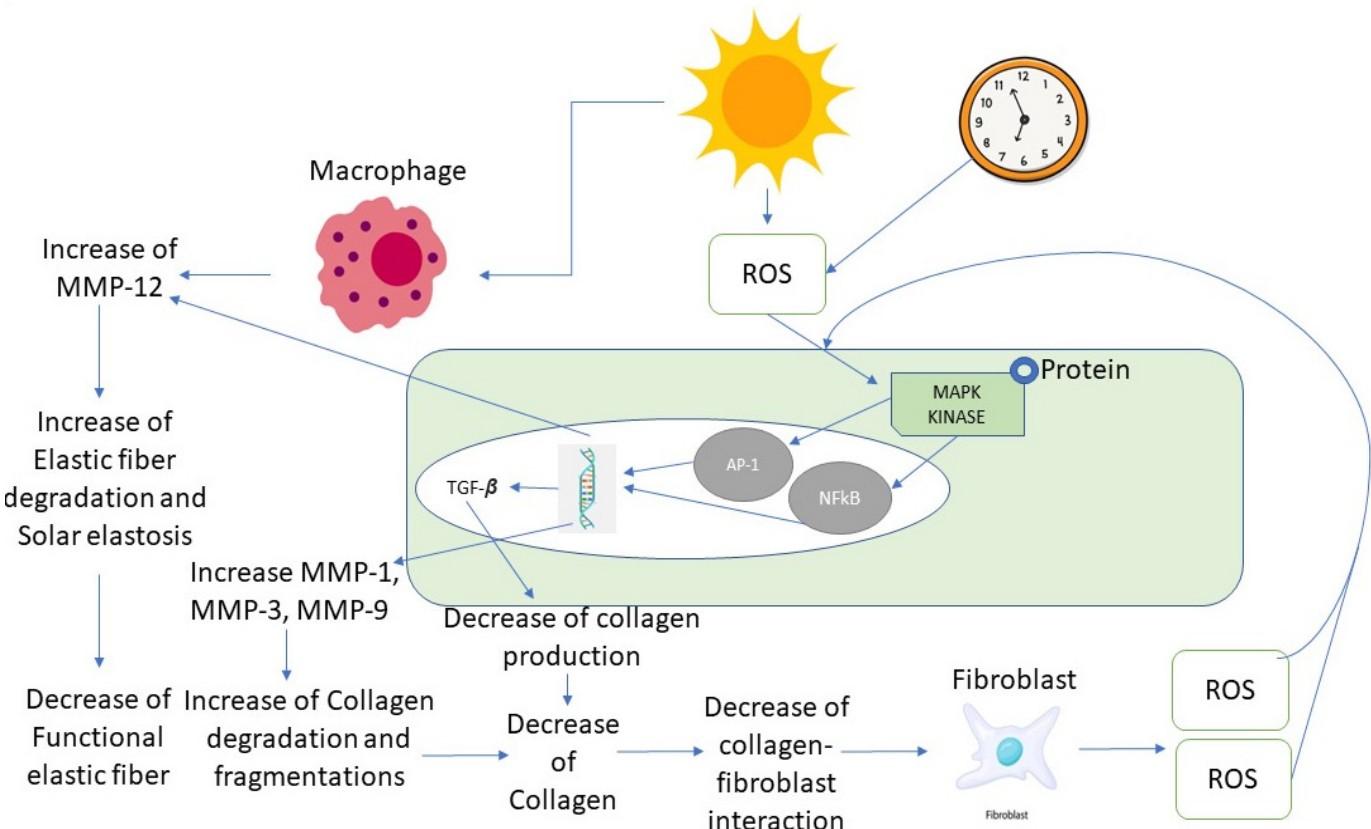

**Figure 1.** Skin aging process [Reproduced from [15]].

Wrinkles develop when the elasticity and strength of our skin are hindered because of the damage to collagen, elastin, and reticular fibers found in the dermis (the middle layer of the skin). The outermost layer, the epidermis, has five layers with the superficial layer termed the stratum corneum [16].

Collagen, elastin, and glycosaminoglycans (GAGs) are the three major targets for designing anti-aging strategies [17]. The extracellular matrix (ECM), a macromolecular

network, contains collagen, elastin, hyaluronic acid (HA), proteoglycan, etc. which maintain the cell regulation, growth, and differentiation needed for youthful-looking skin. Excessive degradation of ECM leads to skin aging due to the deregulation of Matrix Metalloproteinases (MMPs). ECM is a zinc-dependent endopeptidase regulated by the specific endogenous tissue inhibitors of metalloproteinases (TIMPs) [18–21]. Reduced levels of TIMP-1 may elevate the amount of MMPs, thus accelerating skin aging. Additionally, the degradation of TIMP-1 is prominently caused by Reactive Oxygen Species (ROS) [20,22–25]. The mitogen-activated protein kinase (MAPK) family is activated by ROS, and this activation further activates the transcription factor activator protein 1 (AP-1). As a result, the transcriptional regulation is maintained of MMP-1, MMP-3, MMp-9, and MMP-12. The family members of MMPs such as MMP-1, MMP-2, MMP-3, MMP-9, MMP-10, MMP-11, MMP-13, MMP-17, MMP-26, and MMP-27 have been found to be responsible for skin aging [26–30]. The AP-1 activation thus causes collagen degradation leading to aged skin. In addition, UV ray also activates nuclear factor-κB (NF-κB), which plays a role in photoaging by increasing the levels of MMP-1 and MMP-3 causing collagen degradation and leading to deep wrinkles. Both AP-1 and NF-κB elevate the level of matrix metalloproteinase (MMP) expression, and that hinders the growth of transforming growth factor-β (TGF-β) signaling that leads to collagen fragmentation and a decreased amount of collagen synthesis. Moreover, MMP-12 obtained from macrophages and fibroblasts causes solar elastosis and reduces the functional elastic fibers in our skin [31–37].

Skin is the reflection of our health and wellness as well as our psycho-physical status [38]. As we age, the skin becomes rougher, itchy, dehydrated, thin, and less elastic. Aged skin thus plays an important role in our mental health. Skin that does not feel good or look good can be burdensome emotionally and can cause emotional distress like anxiety, depression, and social withdrawal, making it important for aging people to prevent skin wrinkles [16,39]. While it is necessary to design anti-wrinkle agents that successfully target the skin aging mechanism (see more in Figure 2), it is also of utmost importance to address the need for enhancing the administration of anti-wrinkle therapies.

Microneedles (MNs) in the cosmeceutical treatment of wrinkles are one the most promising approaches because of their minimally invasive system; in addition, they can easily overcome the drawbacks of conventional treatments [40–42]. There are many techniques that have been established to date to fabricate these MNs such as thermal inkjet printing, stereolithography, micro-electromechanical systems, etc. [43–45].

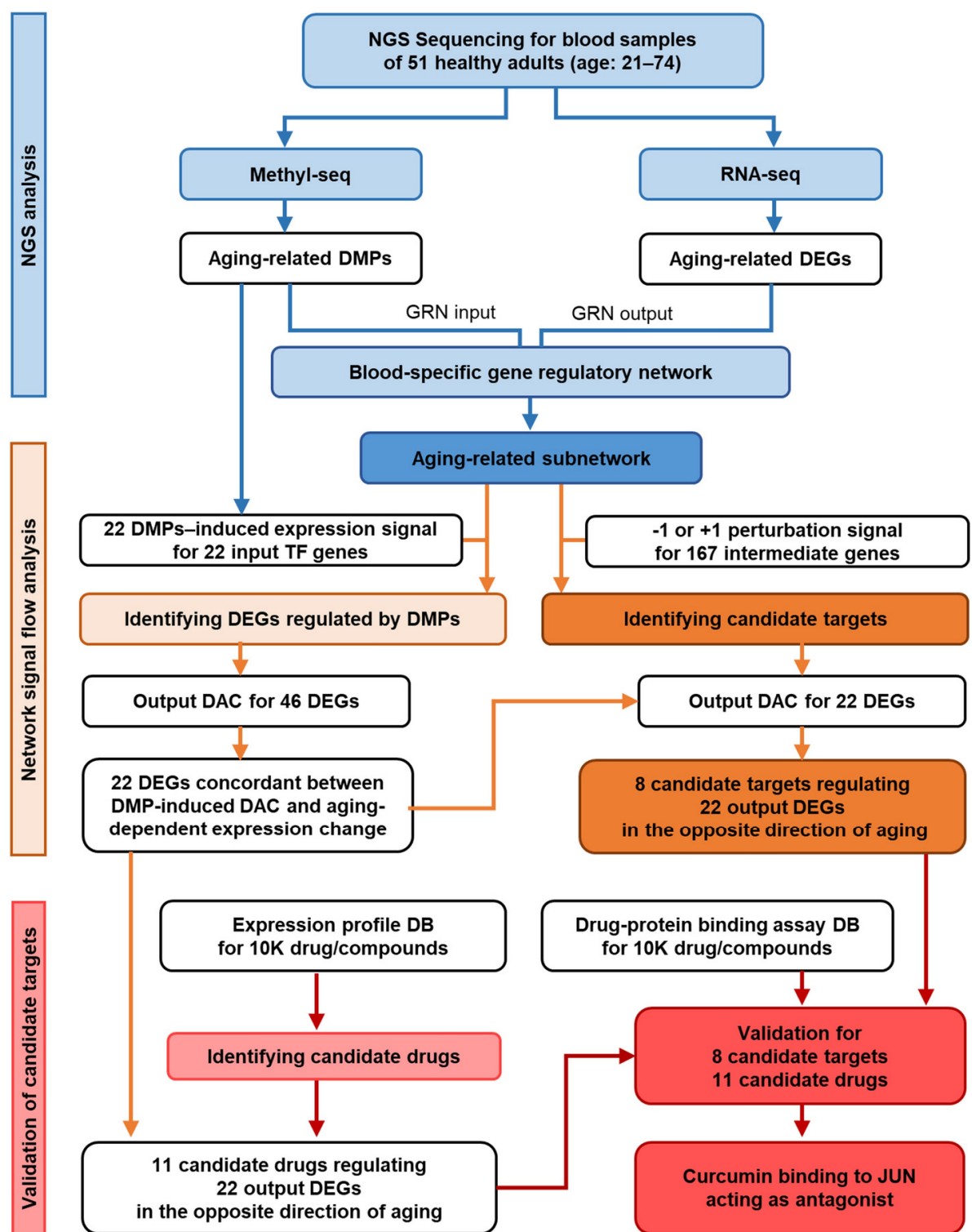

**Figure 2.** Identification process for anti−wrinkle drug molecular targets using AI [46].

*Microneedle-Mediated Anti−Wrinkle Therapy*

MNs are widely used in transdermal drug delivery because of their safe, painless, and non-invasive drug administration. Their construction mimics the needles with a very narrow diameter and length, which penetrate through the stratum corneum and move into the layers of the dermis without affecting blood vessels or the sensory neurons of pain [43,47]. MNs also help the drug get absorbed in the ISF and ease the ISF to diffuse

through the skin painlessly [48,49]. Thus, it is one of the painless dosage forms that can be administered by patients without supervision [50]. The greatest barrier in transdermal drug delivery is the route of administration, which is stratum corneum (SC). SC is the initial protective layer of the skin, which hinders the absorption of drugs into the skin. The microneedle technique can surpass the SC due to its micron-sized needle with a solid base size ranging from 25–2000 μm that can easily be inserted into the stratum corneum of the skin [51], and thus this trait is considered to be one of the major advantages of the microneedle technique [52]. Moreover, it has become a technique of interest for transdermal immunotherapy and can ensure the delivery of active molecules such as antibodies, allergens, and other agents of therapeutic advantage directly to the skin [50]. In a recent case study, coated MN has been used to mitigate the symptoms of allergic rhinitis by regulating IgE in mice model through the skin. Two Phase I trials are currently being conducted to assess the safety and immunological response to allergy immunotherapy compared to the subcutaneous immunotherapy (SCIT) [53]. Specific skin depth is also targeted by the microneedle product and an in vitro skin test also proved the successful piercing of MNs in the layers of skin or epidermis. The technique also causes minimal invasiveness while delivering the active drug molecules to the skin (see Figure 3). Microneedle therapy also eliminates needle administration, making it safer than other techniques such as dermal fillers and Dysport injections by diminishing needle stick injury or needle-induced infections. The administration can also be done without the supervision of qualified personnel, which reduces the overall healthcare cost [54,55].

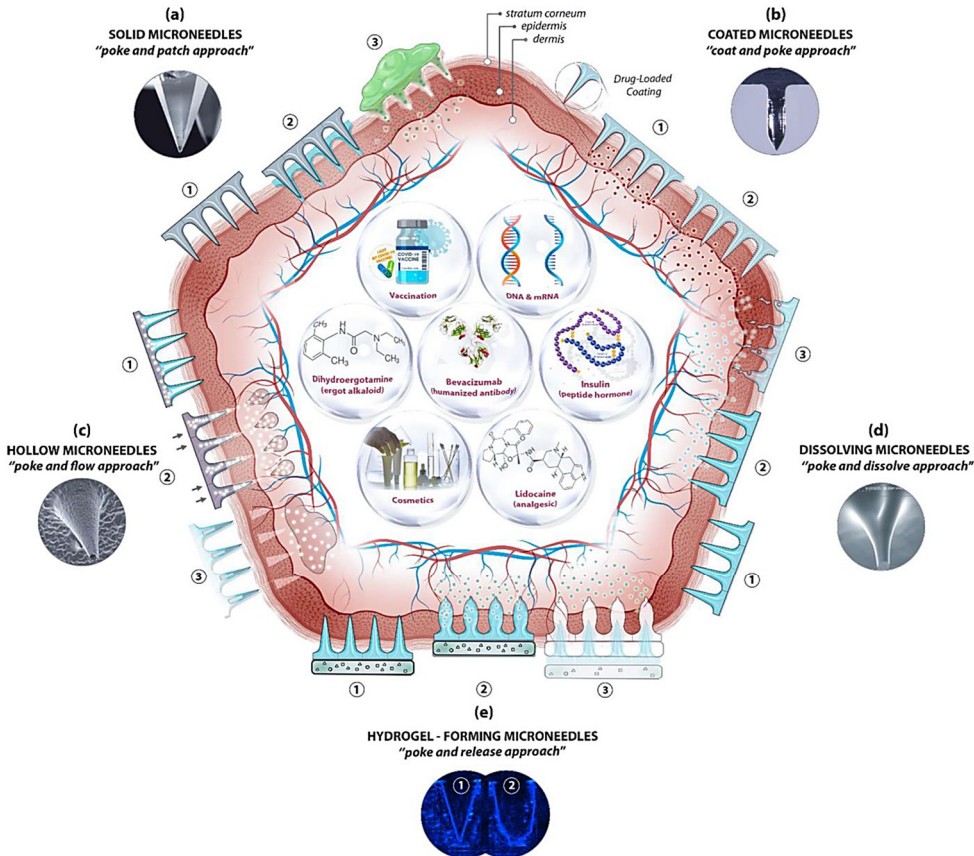

**Figure 3.** Drug delivery via different types of MNs: (**a**) solid MNs, (**b**) coated MNs, (**c**) hollow MNs, (**d**) dissolving MNs, and (**e**) hydrogel-forming MNs. The step-by-step process of each delivery approach is numbered from 1 to 3 [56].

In the cosmetic field, numerous microneedle-mediated treatments have been employed to mitigate the problem of skin wrinkles. Table 1 below summarizes the data from some studies conducted in the last few decades on such treatments.

**Table 1.** Microneedle-mediated treatment studies related to skin wrinkles in the last few decades.

| Sl no. | Type of MN Used | Active Ingredient | Method | Result | Limitation of the Study | References |
|---|---|---|---|---|---|---|
| 1. | Soluble microneedle patch | HA 16.7% (w/w) | Double-blind clinical trial done for 10 to 12 weeks on 84 Korean females with Crow's feet and evaluation of other parameters e.g., dermal density, elasticity, and hydration | Effective reduction of epidermal wrinkle after 8 weeks. Other parameters like dermal density, elasticity, and hydration had increased | The clinical test has been done only for Korean skin types; Discontinuation effects were not properly mentioned, giving rise to durability concerns | [57] |
| 2. | Microneedle eye patch | Acetyl hexapeptide-3 (AHP-3) (500 μL of 10 % w/v), Polydimethyl-siloxane (PDMS), PrestoBlue® | PMNP implemented increases the delivery of AHP-3 through human cadaver skin of Caucasian female, as compared to FMNP and intact skin to analyze the skin wrinkle reduction. | Efficiently penetrated skin very fast compared to other MNs through the first application; cumulative permeation of AHP-3 across human cadaver skin was approx. ~90× higher compared to intact skin; and ~45× higher than FMNP treated skin | Dose adjustment based on an individual patient is difficult; some allergic reactions can be found in small peptides (rare case); high-cost issues; risky for people with skin cancer | [58] |
| 3. | Dissolvable microneedle | Adenosine incorporated low and high MW HA dissolving MN array | Clinical efficacy and safety tests were executed for 12 weeks long upon 3 Korean females with periorbital wrinkles. Ad-HMN and Ad-LMN, both type of arrays were applied in every three days, in the evening, for 8 weeks. Then, skin wrinkling, dermal density, and elasticity were measured. | Both of the groups showed statistically significant efficacy for most parameters; the adenosine incorporated high MW array had better effect on the mean depth of wrinkles, maximum depth of wider wrinkles, dermal density, and skin elasticity than the adenosine incorporated low MW array | Fabrication of dissolvable MNs with high molecular weight hyaluronic acid is a challenge due to the high viscosity of HMW HA | [54,59,60] |

**Table 1.** *Cont.*

| Sl no. | Type of MN Used | Active Ingredient | Method | Result | Limitation of the Study | References |
|---|---|---|---|---|---|---|
| 4. | Dissolvable microneedle patch | Horse-oil and Adenosine loaded dissolving microneedle patch (HOS-Ad-DMN patch) | In-vitro analysis was conducted to ensure successful delivery of the active ingredients with a specific composition. Clinical efficacy and safety tests were conducted on the lateral canthus of 20 Korean women to assess and compare the efficacy of HOS-Ad-DMN patches with that of Ad-DMN patches. | Compared with Ad-DMN patches, HOS-Ad-DMN patches significantly improved skin elasticity, hydration, dermal density, and wrinkles | The active compounds for DMNs have been limited to hydrophilic compounds, because most DMNs are made using hydrophilic polymers for the backbone matrix; delivery of lipophilic compounds via DMNs is a challenge because of the difficulty of attaining a homogenous viscous solution for DMN fabrication; the fabrication with this heterogeneous solution produced uneven DMN morphologies and irregular encapsulation of the active compounds | [61] |
| 5. | Dissolving, detachable microneedle technology | Hyaluronic and ferulic acids | 650 micro-needles, which dissolve in 25 min of exposure, were tested on 82 subjects in a randomized split-phase study. Effectiveness was assessed at 6 weeks. | Demonstrated a significant reduction in the average roughness index, with a steady decrease in puffiness of the application area, increased elasticity and reduced severity of epidermal wrinkles | Additional studies of soluble MNs are required to fully assess the amount and distribution area of the injected hyaluronic acid and other active components | [62] |
| 6. | 3D printed personalised microneedle | Acetyl-hexapeptide 3 (AHP-3), polyethylene glycol diacrylate (PEGDA) and vinyl pyrrolidone (VP) | AHP-3 loaded MNs patches were loaded with polyethylene glycol diacrylate (PEGDA) and vinyl pyrrolidone (VP) and tested on Human cadaver dermatomed skin of a 30-year-old, Chinese Male. | Significant absorbance rate observed with vinyl pyrrolidone (VP) showing highest mechanical rate among other resins; the resin polymers and the 3D printing process showed good biocompatibility on human skin | Study conducted on one subject only; microneedle patches may not be of sufficient flexibility to account for the minor indentations or variations of the skin | [63] |

Dissolvable MNs (DMNs) arrays have recently been widely investigated in the area of non-cosmeceuticals in relation to topical cosmetics, with an emphasis on active ingredient delivery [51]. DMNs have been developed in the field of cosmetics to deliver active com-

pounds such as retinyl retinoate, ascorbic acid, 4-n-butylresorcinol, and epidermal growth factor through the skin in a way that is minimally invasive for wrinkle reduction and skin depigmentation [64,65]. For the treatment of epidermal wrinkles, various formulations are being tested, and some are already on the market. There are peptides and nucleoside-dependent MN delivery systems that have been shown to be effective in treating skin conditions and improving skin texture and can be personalized based on patient compliance [66].

For the treatment of epidermal wrinkles and improving skin textures, various formulations are being tested as previously discussed. However, there are some disadvantages to MNs, such as the diameter and length that sometimes cannot be adjusted to avoid the unpleasant pain induced by microneedle injection. Most of the MNs face problems with high MW compound loading, poor penetration of peptides into the skin, and being subjected to metabolic degradation by peptides that minimize product stability which, in turn, lowers patient compliance [67,68].

For instance, despite having a large molecular mass (889 Da) and a hydrophilic character, AHP-3's skin permeability is frequently restricted, which limits its efficacy and safety as an anti-wrinkle therapy. For this reason, it cannot reach the site of action in an accurate manner after the delivery [63].

Using a transdermal medication delivery system with MNs assists AHP-3 to penetrate the skin more effectively through the tiny pores. The majority of the studies conducted have been done using solid, coated, and dissolvable MNs [69]. A recent study has incorporated a small peptide with hyaluronic acid through dissolvable MNs. This includes the delivery of arginine/lysine polypeptide, acetyl octapeptide-3, and hyaluronic acid to 20 subjected women, >41 age, for 12 weeks. The study found an approximately 15% improvement in hydration, 13% improvement in dermal thickness, and 14% improvement in the density of the skin after 8 weeks. In addition, the process ensured dermal elasticity and overall hydration of the skin. Also, the process ensured dermal elasticity and overall hydration of the skin [70].

Another research has been done through personalized microneedles using the 3D printing technique. Here the active ingredient was AHP-3, and the experiment was done on human dermatome cadaver skin to analyze the MN geometry. The optimized MN geometry was then used to construct a personalized microneedle patch (PMNP), which had shown improved performance. When compared to commercial flexible microneedle patch (FMNP), the delivery of AHP-3 was up to $90\times$ better with PMNP [63,71]. However, this type of MN sometimes leads to ineffective delivery to the target site due to its short duration of action. For AHP-3-like compounds, dissolvable MNs cannot bear a sustainable effect, which can result in less efficacy and need a relatively higher dosage of therapy. Moreover, due to higher molecular weight, it needs more effective technology [72].

Considering all of the above points, in this paper we have focused on discussing the bioinspired 3D hollow microneedles (HMNs), treatment scopes, comparative discussions among MNs, and future aspects of HMNs for treating wrinkles.

## 2. Scope of Treatment for Skin Wrinkles

The challenges that researchers face while designing the required anti-wrinkle treatment therapy largely involve finding the right antioxidants like vitamin C and E or a combination of antioxidative enzymes such as catalases and peroxidase. Apart from that, it is essential to identify if beta-carotene and vitamin A and E are provided in the right amount for the treatment as it can bring about some unwanted side effects [73,74]. In the case of stem cell therapy, the transplantation of adipose-derived stem cells in the appropriate skin layer has to be ensured. Mistreatment of the transplantation must be avoided to ensure skin rejuvenation and produce necessary growth factors to reduce aging. In the case of hormone replacement therapy for treating skin wrinkles, it works significantly well to slow aging and improve elasticity, but the concentration and extent of the use of artificially produced hormones need to be verified as it can increase the risk of developing breast cancer [74,75]. Telomere modification for treating skin wrinkles induced a high expression of telomerase reverse transcriptase in skin keratinocytes. However, it increased the risks of epidermal carcinogenesis during this time. Apart from these, certain dietary considera-

tions, anti-progeria, and anti-inflammation strategies for treating skin wrinkles need to be considered before coming forward to design an effective anti-wrinkle treatment procedure.

The previous antiwrinkle therapies were ideally poised for the treatments, although they had certain limitations. For instance, botulinum toxin injections are a very efficacious method for treating skin wrinkles intradermally. It is primarily a soft tissue filler, which may consist of components like collagen, fat, and HA [76]. One more drawback of the botulinum toxin injection is that the efficacy of the medication usually takes up to 3–4 months to show its effectiveness [38]. Laser treatment to treat wrinkles uses Er-Yag and $CO_2$ lasers, which take a handsome amount of time to show significant changes. Apart from this, hormone replacement therapy has a higher tendency to bring about an increased risk of breast cancer [77].

With more research in the cosmetic field and the advent of technology, a variety of techniques have become available for treating wrinkles, ranging from topical creams and serums to numerous surgical procedures to smoothen out wrinkles and fine lines. One of the first vitamins approved by the FDA to treat wrinkles is vitamin A. Retinol, retinal, and retinoic acid, which have the same biological features as vitamin A, slow down the aging process and are now frequently used in anti-wrinkle treatment [78,79]. The formulation used for their anti-wrinkle effects include alpha hydroxy acids (AHAs), poly-AHAs, complex poly-AHAs, retinoids, fish polysaccharides, anti-enzymatic agents, antioxidants, such as ascorbic acid, pycnogenol, ursolic acid, vegetable isoflavones, vitamin E, coenzyme Q10, lipoic acid, resveratrol, l-carnosine, and taurine, as well as agaricic acid and various plant extracts [80]. Even though vitamin A and retinoids are commonly used in anti-wrinkle therapy, the use of vitamin A and its derivatives in pharmaceutical and cosmetic products has been limited due to their instability and irritant properties [78].

Advanced anti-wrinkle therapies include the use of adenosine-incorporated high and low molecular weight (MW) HA-dissolvable MN arrays, which have been proven to be clinically efficacious [54]. As vehicles, synthetic peptides have been incorporated into the formulations for the desired results. Some of the commonly used peptides to treat skin wrinkles include palmitoyl tetrapeptide-7, palmitoyl oligopeptide, and acetyl hexapeptide-8 [81]. The major anti-wrinkle agents that have gained significant attention in the cosmetic industry have been summarized in Table 2.

**Table 2.** Current anti-wrinkle agents in use.

| Reagent Used in Formulation | Nature | Function | References |
|---|---|---|---|
| Retinyl retinoate | Lipophilic | Photostable; have lower toxicity and greater skin rejuvenation than retinol; effective in treating periorbital wrinkles | [82] |
| Ascorbic acid | Hydrophilic | Acts as an antioxidant; effective in treating photo-aging | [83] |
| Hyaluronic acid | Hydrophilic polysaccharide | Increases skin moisture and reduces the appearance of fine lines and wrinkles by improving collagen and elastin stimulation | [84–86] |
| Adenosine | Amino acid | Effective in treating crow's feet and frown lines by skin density, elasticity and hydration | [87,88] |
| Horse oil | Lipophilic | Restores stratum corneum and imparts skin-moisturizing effects | [89] |
| Acetyl-hexapeptide 3 (AHP-3) | Small peptide | Decreases the anisotropy of skin to help in treating skin wrinkles | [90] |
| Epidermal growth factor | Small water-soluble polypeptide | Effective in treating periorbital wrinkles | [85,91] |
| Ferulic acid | Hydrophilic | Antioxidant that helps to maintain the skin's smooth morphology by reducing the development of fine lines, spots, and wrinkles | [92] |
| Ceramide | Lipid | Restores skin moisture, balances skin pH, reduces skin wrinkles and Trans Epidermal Water loss (TEWL) | [93] |
| Niacinamide | Vitamin B3 | Improves skin elasticity, reduces skin wrinkles and fine lines, decreases hyperpigmentation and skin sallowness, minimizes pore size, and eases skin inflammation | [94] |

### 3. The Potential of HMNs to Treat Skin Wrinkles

HMN therapy can bypass the protective layer of the skin, enabling the direct delivery of therapeutics into the skin [47,95]. HMNs can permeate the skin more effectively than other techniques including MN patches and electroporation patches [96]. There are void spaces in HMNs filled with solution and holes at the tip that deliver the drug after skin penetration. By increasing MNs bore, it increases the flow rate, but the mechanical strength and sharpness of HMNs decrease [97]. Their needles are micrometric in size and that makes the manufacturing process difficult and expensive [98]. Roxhed et al. published an article in 2008, where they discussed a good number of techniques in HMN preparation, with a view to finding a feasible way of fabrication [99]. However, given its micrometric size, the patient's acceptance is higher than with a conventional injection [98]. Due to its high penetration and permeability, when the void is incorporated with anti-wrinkle medications, it will release the therapeutic agents readily and easily into the epidermis directly facilitating skin rejuvenation [97]. In October 2008, the HMN array got patented in the USA (US-2011213335-A1) by a group of researchers as a transdermal drug delivery device [100].

Moreover, recent developments in HMNs show that they can deliver a wide range of drug molecules. HMNs can be the safest option for delivery in the case of a small peptide as it has a higher delivery capacity than other MNs [72]. In 2013, Fabbrocini et al. conducted an experiment on 60 patients, skin prototype of I to VI, where they delivered collagen via MNs percutaneously for three months at a monthly interval. They observed a reduction in the degree of irregularity of skin texture with an average reduction of 31%. Moreover, no short- or long-term dyschromia had been observed among the participants in their study [101]. Passive diffusion is the easiest technique of drug administration employing HMNs. Faster transport rates by pressure-driven flow or diffusion have consistently been accomplished because the passive diffusion rate in thick tissues is remarkably low. Therefore, small peptide delivery can bring more efficacious results if incorporated into HMNs [102]. The U.S. Food and Drug Administration (FDA) granted approval for Micronjet®, an HMN designed by Nanopass Technologies, for use in intradermal delivery of any drug substance which is approved to be administered via transdermal route [72]. In addition, commercially available FDA approved Dermaroller® is being extensively used in cosmeceutics [103,104]. In 2017, Oyunsaikhan et al. conducted an experimental work on 19 healthy women, aged 43–48 years, with scores ranging between 2–4 on the baseline facial fine wrinkle. They were treated with a platelet rich plasma (PRP)-incorporated Dermaroller® for 8 weeks at a 4-week interval and a significant change in dermal fibers, epidermal thickness, papillas, and skin glands were observed, which led the researchers to conclude that Dermaroller-based treatment is a promising novel method of facial rejuvenation compared to others [105]. In 2020, an experiment was conducted by Subburaj et al. on the treatment of vitiligo. There was a total of 36 participants having at least three vitiligo patches in the same anatomic region with a minimum lesional stability of one year. The patients were treated via Dermaroller® followed by noncultured epidermal cell suspension (NCES), which is a well-established surgical treatment modality for stable vitiligo. Patients were followed up at 4, 8, and 12 weeks, and assessments of the extent and pattern of repigmentation, color match, and patient satisfaction were done. It was observed that Dermaroller® had poor repigmentation outcomes compared to others (dermabrasion and cryoblister) [106].

Other drug molecules that have a high molecular weight can be effectively delivered by HMNs directly to the skin. The high molecular weight compound includes proteins, vaccines, and oligonucleotides [97]. V-Go® is an FDA-approved MN patch manufactured by Valeritas and is used to deliver insulin [107]. HMNs can be useful in the delivery of immunological products as well because they can deliver many dendritic cells, macrophages, lymphocytes, and mast cells into the layer of the skin. A microneedle accumulates a great number of dendritic cells, macrophages, lymphocytes, and mast cells in the epidermal layer. As a result, it improves and expands immune-reprogramming strategies [47]. Silicon-based HMNs show more efficacy because of excellent biocompatibility and mechanical properties

than the other types of HMNs [108,109]. Silicon has enough mechanical strength that helps in easy skin penetration. They can be manufactured with a small and sharp tip, which will increase the efficacy of skin insertion [110]. Currently, polymeric HMNs have gained attention because polymers are biocompatible and biodegradable. Polymeric MNs do not leave any sharp wastes like silicon HMNs. However, using polymers can be challenging as they tend to be soft and can have lower mechanical strength compared to silicon hollow MNs [111,112].

Despite the advantages of HMNs, if the fabrication of hollow microneedles is not precise, uncontrolled drug release from the needles may occur. As a result, the micro-channels created by HMNs for drug penetration may become blocked [113]. Thus, the fabrication technique for 3D printing HMNs needs to be optimized as well.

## 4. Fabrication of 3D Printed HMNs

Yeung et al. conducted a study that showed the utilization of a 3D printed microneedle of hollow design where the fine-tipped syringe-shaped microneedle had undergone a thorough assessment study where rhodamine B, fluorescein iso-thiocyanate, and methylene blue were used through the device. These molecules were considered to have a similar weight to model drug substances, similar diffusion profile, and emission peaks [114]. The study provided an idea of fluorescence depth and analysis due to the fluorescence produced by the particles, and also that molecules responsible to reduce skin wrinkles like antioxidants, anti-UV agents, anti-inflammatory agents, etc., can be optimized through 3D printed hollow therapy to improve skin hydration (e.g., hyaluronic acid) and elasticity (e.g., retinoids) and reduce skin wrinkles.

3D printed HMN therapy has the capacity of regulating the dosage and the dose intervals of the drug provided. This allows for reducing drug toxicity over a specific portion of the skin to a great extent. This also allows to bring about a change in the efficacy of the drug since only the required amount of drug can be allowed. This method provides a more permeable response to create a drug-induced response, and thus allows a higher potency. Having all these advantages at hand, skin rejuvenation can be ensured by reducing the adverse effects of vitamin A/E, various hormones, retinoids, hyaluronic acids, etc. [74]. Furthermore, this method allows a higher rate of dose adjustment, which makes it more effective to avoid adverse effects like breast cancer.

Dissolvable MNs fail to provide prolonged drug release, which can be overcome using 3D printed HMNs. State-of-the-art, bio-responsive 3D printed MNs can be manufactured to meet much higher standards than those possible with conventional MNs [115].

There are some advanced techniques to manufacture 3D printed HMNs. Researchers have fabricated hollow and open-channel MNs using 2PP-two-photon polymerization-directed laser scanning printing. Using 2PP-3D printing, current research has created MNA master molds with pointed MNs (tip radius of 500 nm) to make it easier to develop multicomponent MNAs through additional molding steps. Therefore, 2PP-based 3D laser printing has the capability to be a flexible method for the precise and repeatable production of high MNAs with sharp MNs [116]. Trautman and colleagues reported the printing of another common hollow microneedle using the TPP method. Their study produced a microneedle-based microfluidic device that may have a wide range of intriguing biological applications, including precise, non-invasive medication administration of HMNs [117]. In addition to producing complex micro- or, nano-objects with greatly increased geometry and resolution control, TPP is also able to significantly lower the overall cost of the manu-facturing process. Proper maintenance of facilities and equipment is frequently related to the post-etching stage of lithography-based processes [53].

On the other hand, stereolithography can be used to construct 3D printed MNs with the desirable versatility and complexity to create the ideal microfluidic and micro-scale MNs needed for optimum transdermal drug delivery as compared to the conventional dissolvable ones and HMNs, which do not provide much freedom and flexibility when developing the MNs [114]. The 3D printed hollow microneedle approach has turned out

to be a very accurate and reliable means of method. In a study done on guinea pigs on perilymph sampling from the inner ear for the purpose of proteomic analysis, the 3D printed HMNs after perforation and sampling showed a very negligible amount of needle break/bending, which was tested in scanning electron microscope after sampling [118]. This sampling method gives an idea of the reliability of 3D printed HMNs despite being a complex process. Another recent study has also shown a hollow microneedle patch system as a form of a platform for drug delivery with a MEMS (Microelectromechanical system), which is another testament to the versatility of 3D HMNs [45].

Furthermore, stereolithography (SLA) printers have significantly higher efficiency than conventional two-photon polymerization (TPP) printers and can print submillimeter/centimeter-scale elements with larger construct volumes, greater height, and faster print durations, such as 15 min for a comparable 6 mm in length [119,120]. Transdermal medication delivery devices can be quickly prototyped and customized to satisfy the demands of a specific patient and combine this with the lower price and lightness of commercially accessible SLA printers [121].

There is another technique that can be highly efficient for the fabrication of HMN known as Fused Deposition Modeling (FDM). This technique provides greater resolution than SLA printers [122]. This can be applicable to all active ingredients by using different filaments in a single process with adequate thermal stability. Furthermore, it serves a huge operation scheme with inexpensive manufacturing of 3D printed hollow MN with Vero Clear printing material and greater geometric features [53,123].

## 5. Techniques Involved in 3D Printed HMNs

It has been shown that metal can be electroplated or electroless plated onto positive or negative MN molds to create HMNs using 3D laser cutting or laser ablation [124]. In addition, laser micromachining has been used to make HMNs (400–800 m in length) by drilling trenches in dissolvable molds to form lumens in the microneedle structures. An infrared laser can be used to cut through metal or polymeric sheets, as well as conical wire, to form HMNs. Computer-aided design (CAD) software allows for the creation of HMNs with the desired shape, geometry, and dimensions. After the needle has been cut with a laser to its desired profile, it is soaked in hot water to remove any remaining debris and then bent perpendicular to the base plane. Subsequently, MN are electropolished, rinsed, and dried with compressed air, which helps to minimize the thickness of MN and sharpen the points. This production technique can be used to create both two-dimensional arrays of metallic MN and single rows of MN with a variety of shapes [125–128]. By employing an excimer laser to drill holes of the required geometry into polyimide or polyethylene terephthalate polymer sheets, a group of researchers successfully manufactured metallic HMN with tapered walls (without the need of master polymeric molds). Electroplating metallic HMNs with tapered walls required the production of tapered holes bored all the way through [129,130]. Micromolding of polyethylene terephthalate with an ultraviolet laser was achieved by Davis et al. via a modified—LIGA—technique. A nickel coating was electrodeposited onto a sputter-deposited seed layer, and then 16 metal MN (500 nm long, 75 nm in diameter at the tip) were freed from the polymeric molds using selective etching. When subjected to mechanical testing, these MN were shown to be robust enough to pierce living tissue without fracturing [131]. Table 3 summarizes the materials commonly used in HMN fabrication along with the fabrication technique.

**Table 3.** Materials and fabrication technique of Hollow MNs (Cárcamo-Martínez et al., 2021).

| Material Group | Material Subtype | Fabrication Technique | References |
|---|---|---|---|
| Silicon | Monocrystalline, Polycrystalline silicon | Etching, lithography, deep reactive ion etching (DRIE) | [108,132,133] |
| Metal | Stainless steel, titanium, palladium, palladium-cobalt alloys, nickel | Laser cutting, laser ablation, etching, electropolishing, lithography, microstereolithography, deep reactive ion SU-8 UV-LIGA | [124] |
| | Nickel | Moulding, Nickel electrocoating and Electropolishing | [64] |
| | Pt-Metallic glasses | Thermoplastic Drawing | [134] |
| Carbon | Glassy carbon | Conventional Micro-electromechanical system (CMEMS) process | [135] |
| Borosilicate | Fire-polished borosilicate glass | Pulling pipettes | [136] |
| Miscellaneous | Polycarbonate | Injection micromolding | [137] |

Transdermal microneedle patches are a potential method for delivering several therapeutic ingredients into the skin. To act as a replacement for conventional hypodermic needles, MNs must puncture the stratum corneum of the human skin without tearing or bending. This guarantees that the material is discharged at the specified time and location [138,139]. Consequently, the capacity of MN patches to adequately puncture the skin is an essential necessity. Despite the task's apparent simplicity, a number of criteria, including needle length, geometry, thickness and point radii, base diameter, needle saturation, and MN material, are responsible for proper MN insertion [140,141].

Such parameters may be modified to produce the optimal amount of MN penetration necessary for effective drug delivery or specific site targeting. Therefore, modifying the MN design, geometry, and manufacturing material can affect the needles' capacity to resist the skin's inherent elasticity and pierce the skin, all of which can be adjusted to enhance drug delivery on a case-by-case basis [142].

Moreover, a number of key design aspects of the MNs highly influence the therapeutic efficacy of the active agent. The composition of fabrication influences the strength, rate of release, and drug loading capacity of the MNs [143]. The height of the needles plays a vital role in every determining aspect of the efficacy [144]. The patch area determines the targeted area, cells, and drug loading capacity [145].

Nevertheless, these techniques of manufacturing MNs using various expensive elements can be of higher cost sometimes, specifically for metal MN arrays for large batch size production or to produce other MNs such as solid, coated, or dissolvable MN [146]. However, in the case of HMN, new methods can be incorporated to make it more cost-effective [97]. Researchers suggest a high-fidelity replication technique with a high potential for massive yields to address the existing challenges. In order to create a metal HMN array template, the tips of traditionally manufactured hypodermic needles are used, and this template is the basis for creating an elastic polydimethylsiloxane stamp (PDMS). Subsequently, it replicates the hollow structure of the needle arrays using the PDMS stamp, accomplishing the inexpensive and mass production of HMN arrays made of various materials [147,148]. This construction process allows for an easy adjustment of the traditionally manufactured hypodermic needle arrays like needle size, density, height, and area. Concurrency was possible since the replicating processes showed great consistency throughout the replicated HMN arrays. Each microneedle array is expected to cost less than 20 cents due to the repli-

cation technology and polymer material, which makes them very affordable and possibly mass-producible [149,150].

## 6. Future Directions

In order for 3D printed MNs to be effective in reducing the appearance of wrinkles, they must be able to reduce the fat and increase the collagen production in the dermis layer of the skin. Hollow MNs promise to be successful to treat skin wrinkles and other skin conditions, but there are a few challenges with hollow MNs which must be overcome in future modifications. The drug flow rate within the hollow MNs and the sharpness of the MNs must be balanced and optimized in a way that maximizing the flow rate does not lower the sharpness of the MN during the fabrication [97]. The total amount of drug released and the rate of release from the skin can be adjusted using an appropriate formulation technique [91]. The biodegradability of hollow MNs made of hyaluronic acid is lost when it is covered with parylene. As a result, the hyaluronic acid hollow MNs can be coated with PLA in the future, which is a biodegradable material with a slower degradation rate than the hyaluronic acid [151]. Moreover, the micrometric size of the hollow needle increases the cost of the manufacturing process. For better patient compliance, the fabrication of needle size can be optimized to make it more economical [98]. Figure 4 shows an illustration of the clinical trial process of HMN in the treatment of wrinkles.

Micromolding, electrodeposition, mold-based etching, and/or solvent casting can be used to make HMNs [132] in the future. 3D printing is also a viable manufacturing technique to develop hollow MN which will allow for both micro-molding and direct fabrication of patches using two-photon polymerizations (TPP), a comprehensive photopolymerization printing technique. TPP has already been used to create a patch that combines a hollow MN and a reservoir for both transdermal and implantable systems. However, high prices, longer printing times, and limited printing volumes obstruct mass manufacturing, raising doubts about the technology's potential for widespread clinical use [114]. Due to the micro height and lesser tip sharpness of the MN array, the administration process is rendered to be minimally invasive [151]. Once fabrication is done, Scanning Electron Microscopy (SEM) is to be used. Scanning electron microscopes serve as a medium to provide high-resolution imaging of the microneedle array. It is merely used for the investigation of the interface that the microneedle arrays provide and checks the dimension of the patch system as it helps in sufficient resolution and depth focus. This in turn allows the researchers to analyze whether the desired shape and length have been achieved [152]. Next, Optical Coherence Tomography (OCT) can be applied for in-depth imaging to determine the interaction between the porcine skin and the microneedle array. It is said to be sensitive to various refractive indices and, thus, resolves with high-resolution translucid samples. It also uses focused ion beam preparations [153]. Finally, in vitro studies and statistical analyses are to be done to evaluate the methods employed and the results obtained.

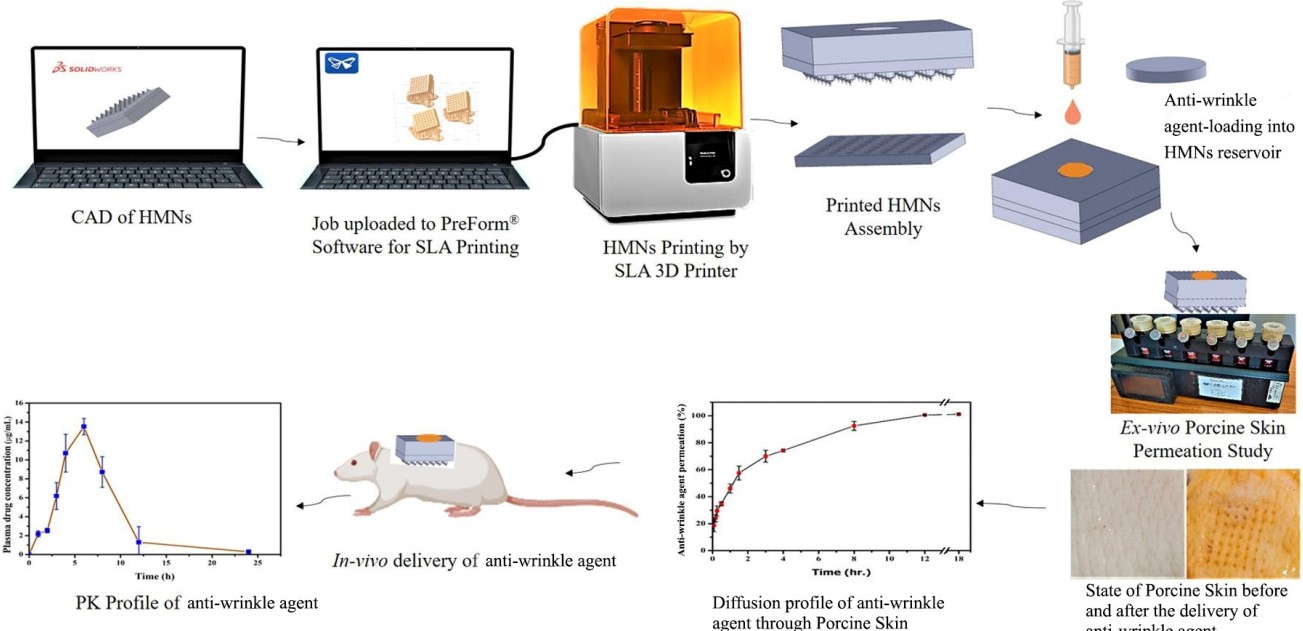

**Figure 4.** A schematic representation of the manufacture of 3D printed HMNs for the administration of anti-wrinkle agents [Adapted with permission from Ref. [121]].

## 7. Discussion

The 3D printed hollow MNs are considered a safe option for the transdermal delivery of drugs. The efficacy rate for drug delivery using 3D printed hollow MNs can be assessed by checking the flow rate of the drug using in situ modulation. Despite the promising potential of 3D printed MNs in general, there are a few drawbacks pertaining to it. The compliance between the elastic moduli of skin and MNs is necessary so that there is no leftover chip or parts in the skin during insertion and no undesirable incidents occur. MNs are very susceptible to fracture during application while going for self-administration. Ultra-fine needles ensure that this incident does not occur and thus no extra supervision from the health care experts may be required. The use of a silicone lubricant may help to ease the pain during insertion [122]. 3D printed MNs have even found their profound use for bio-signal detection. There are two methods, one is the wet-electrode, which is time-consuming and uses gel, and the second one is the dry-electrode, which shows large impedance values. Both processes are very high-cost laboratory methods and require auxiliary powers to function properly. For this reason, removing hair lines or the outermost skin layer is suggested for accurate results. Another solution may be to use electrolytic gel to reduce the impedance, but the long-term use of these gels is harmful and not advised [154–156].

Currently, industries are rethinking the commercialization and scalability of 3D MNs and looking for easier ways for simple and customizable methods for fabrication for generating low-cost 3D printed microneedle production. MNs prepared using the desktop SLA 3D printer are comparatively cheaper than the conventional methods and do not require facilities or expertise for microfabrication. The production of high aspect ratio sharp needles are used to penetrate the skin and the production process uses a simple two-step "print and fill" mold.

The problems related to low aspect ratio and poor tip sharpness could be overcome using this low-cost 3D printed microneedle. This two-step process produces MNs that in-lab researchers would be able to use regardless of having expertise in fabrication technique or those who have no access to specialized equipped labs for the production of microneedle arrays [157].

The production of 3D printed microneedles is done using updated printing technologies such as SLA, TPP, and FDM as previously discussed. 3D printed microneedles are also made by inkjet printing, which ensures that there is a droplet deposition system from a nozzle located at the end of the reservoir itself [44]. These microneedle patches are FDA approved. The entire process is done using the visible light dynamic stereo-lithography method. The 3D printed microneedle industry is said to reach as much as 8 billion because of the high-rise demand by the year 2025 and pharma industries are innovating novel ways to generate cost-effective microneedle production systems that can compete in the market and aid in large-scale production [158]. MHRA also approved personalized medicine for treating skin using transdermal microneedles. Biocompatible and scalable microneedle patches were also approved in the UK as a futuristic administration system [159]. However, there are certain ethical issues governing 3D printing MNs that must be taken into consideration before using them in any animal model cell line. The tests related to in vivo and in vitro with 3D printed MNs must be done while ensuring the animal welfare act and maintaining all the rules and regulations provided by the government.

## 8. Conclusions

MNs provide a methodical tool for the delivery of drugs through the transdermal route in a painless way, exhibiting huge potential in anti-wrinkle therapy. Although it is challenging to engineer microneedle arrays as desired due to their micron-sized structure, this novel technique has become popular for the delivery of drugs. The 3D printing method is an effective manufacturing process with supremacy in the fabrication of personalized medicine and complex structures. Nowadays, this technology is being applied to fabricate microneedle patches that can boost the use of MN arrays in varied areas of science such as pharmaceutics, cosmeceutics, etc. Nevertheless, due to its efficiency and being an attraction for scientists, the pharmaceutical industries are still hesitant to invest in its commercialization. Right now there are a total of 13 microneedle-based products available in the commercial market for cosmetic and some pharmaceutical (drug and vaccine delivery) purposes, and the market for MN-based products is predicted to dilate at a compound annual growth rate of 7.1%. Overall, along with deeper knowledge on different strategies and polymers, the progress in 3D printing technology might facilitate the fabrication process of MNs to efficiently provide antiwrinkle therapy in the future.

**Author Contributions:** Conceptualization, M.J.U. and D.B.D.; resources, M.J.U. and D.B.D.; writing—original draft preparation, J.H., H.I., T.S.P., Z.T.T., S.R., A.I.J.N., A.A. and S.N.A.; writing—review and editing, J.H., M.J.U. and D.B.D.; supervision, M.J.U. and D.B.D.; project administration, M.J.U. and D.B.D.; funding acquisition, M.J.U. and D.B.D. All authors have read and agreed to the published version of the manuscript.

**Funding:** This research received no external funding.

**Institutional Review Board Statement:** Not applicable.

**Informed Consent Statement:** Not applicable.

**Data Availability Statement:** There are no additional raw data for this paper. The paper only uses secondary data from published papers, and all credits for these data have been made via citations and copyright permissions.

**Conflicts of Interest:** The authors declare no conflict of interest.

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
