# Peer review of "3D Printed Hollow Microneedles for Treating Skin Wrinkles Using Different Anti-Wrinkle Agents: A Possible Futuristic Approach"

_cosmetics, doi:10.3390/cosmetics10020041_

Round 1

Reviewer 1 Report

The work "3D printed hollow microneedles for treating skin wrinkles using different anti-wrinkle agents: A possible futuristic approach" represents a novel topic related to topical/transdermal delivery of anti-wrinkle agents. Before acceptance of the paper I request authors to do following corrections in the paper:

1. I suggest authors to add some research findings related to 3D printed hollow microneedles for delivery of anti-wrinkle agents with detailed results (in the form or reproduced Figures/Tables) so that readers can understand their clinical or preclinical efficacy better.

2. Please add the current status of clinical trials related to use of 3D printed hollow microneedles for delivery of anti-wrinkle agent.

3. Please add some patents related to use of 3D printed hollow microneedles for delivery of anti-wrinkle agent for improving visibility of paper.

4. Please remove grammatical and typographical errors from whole the manuscript. 

Author Response

The work "3D printed hollow microneedles for treating skin wrinkles using different anti-wrinkle agents: A possible futuristic approach" represents a novel topic related to topical/transdermal delivery of anti-wrinkle agents. Before acceptance of the paper I request authors to do following corrections to the paper:

  1. I suggest authors to add some research findings related to 3D printed hollow microneedles for delivery of anti-wrinkle agents with detailed results (in the form or reproduced Figures/Tables) so that readers can understand their clinical or preclinical efficacy better.

Answer to the reviewer:

The authors would like to thank the reviewer for this suggestion. As suggested, we have added new information which the reviewer can find in line no. 270-288 and Table 2.

  1. Please add the current status of clinical trials related to use of 3D printed hollow microneedles for delivery of anti-wrinkle agent.

Answer to the reviewer:

The authors would like to thank the reviewer for this suggestion. We have added new information which the reviewer can find in line no. 261-265 and 273-288.

  1. Please add some patents related to use of 3D printed hollow microneedles for delivery of anti-wrinkle agent for improving visibility of paper.

Answer to the reviewer:

We have added the new information which the reviewer can find in line no. 257-258.

  1. Please remove grammatical and typographical errors from whole the manuscript.

Answer to the reviewer:

Thank you, we have revised our entire write-up as suggested by the referee which has clearly improved the quality of the paper.

Reviewer 2 Report

This manuscript reviews research to update the latest knowledge on the potential of bioinspired 3D-printed HMNs for the treatment of skin wrinkles. 

It is an enjoyable review. When I am starting to read the introduction (but applies to the abstract too), I immediately ask: "But why is 3D printed hollow microneedles for treating skin wrinkles using different anti-wrinkle agents so important? WHY does it matter?" I find the answer in the clearly written introduction. The introduction outlines the problem and purpose of the review clearly. The abstract too should hit the reader straight away with the crucial importance of this study, if it does present important information of course. I think the abstract is correctly written. The manuscript is technically correct. The manuscript is well organised and clearly presents the main results in the related field. I believe that the topic is original. It is important and relevant in the fields of health sciences, chemical engineering and the field of pharmaceutical sciences. In particular, the topic is very important for practical applications, especially in cosmetology. Such a review should be helpful in systematising knowledge in this field. The English language should be checked for correctness throughout the text. The manuscript appears to be methodologically correct. It is coherent and describes the issues in the topic well.  The conclusions are very well written and consistent with the evidence and arguments presented. Tables and figures contain all necessary information and are appropriately captioned and clear. Units and abbreviations are clear. The reference materials are well selected and up to date, although I am still missing items from 2019-2022. In conclusion, the review is suggested for publication in Cosmetics.  Overall, the topic is interesting, the approach is appropriate for this journal.

Author Response

This manuscript reviews research to update the latest knowledge on the potential of bioinspired 3D-printed HMNs for the treatment of skin wrinkles. 

It is an enjoyable review. When I am starting to read the introduction (but applies to the abstract too), I immediately ask: "But why is 3D printed hollow microneedles for treating skin wrinkles using different anti-wrinkle agents so important? WHY does it matter?" I find the answer in the clearly written introduction. The introduction outlines the problem and purpose of the review clearly. The abstract too should hit the reader straight away with the crucial importance of this study, if it does present important information of course. I think the abstract is correctly written. The manuscript is technically correct. The manuscript is well organised and clearly presents the main results in the related field. I believe that the topic is original. It is important and relevant in the fields of health sciences, chemical engineering and the field of pharmaceutical sciences. In particular, the topic is very important for practical applications, especially in cosmetology. Such a review should be helpful in systematising knowledge in this field. The English language should be checked for correctness throughout the text. The manuscript appears to be methodologically correct. It is coherent and describes the issues in the topic well.  The conclusions are very well written and consistent with the evidence and arguments presented. Tables and figures contain all necessary information and are appropriately captioned and clear. Units and abbreviations are clear. The reference materials are well selected and up to date, although I am still missing items from 2019-2022. In conclusion, the review is suggested for publication in Cosmetics.  Overall, the topic is interesting, the approach is appropriate for this journal.

Answer to the reviewer:

The authors would like to thank the reviewer for the appreciative comment. We have added some new information from 2019-2022. The reviewer may find them in line no. 270-272 and 280-288.